# [^18^F]-Fludeoxyglucose Positron Emission Tomography/Computed Tomography with Radiomics Analysis in Patients Undergoing Aortic In-Situ Reconstruction with Cryopreserved Allografts

**DOI:** 10.3390/diagnostics12112831

**Published:** 2022-11-17

**Authors:** Raffaella Berchiolli, Lorenzo Torri, Giulia Bertagna, Francesco Canovaro, Roberta Zanca, Francesco Bartoli, Davide Maria Mocellin, Mauro Ferrari, Paola Anna Erba, Nicola Troisi

**Affiliations:** 1Vascular Surgery Unit, Department of Translational Research and New Technologies in Medicine and Surgery, University of Pisa, 56126 Pisa, Italy; 2Unit of Nuclear Medicine, Department of Translational Research and New Technologies in Medicine and Surgery, University of Pisa, 56126 Pisa, Italy

**Keywords:** cryopreserved allograft, radiomics, FDG-PET/CT, aortic in situ reconstruction

## Abstract

Background: The aim of this study was to evaluate the effectiveness of positron emission tomography/computed tomography with [^18^F]-fludeoxyglucose (FDG-PET/CT) and radiomics analysis in detecting differences between the native aorta and the abdominal aortic allograft after the total eradication of infection in patients undergoing infected graft removal and in situ reconstruction with cryopreserved allografts. Methods: Between January 2008 and December 2018, 56 vascular reconstructions with allografts have been performed at our department. The present series included 12 patients undergoing abdominal aortic in situ reconstruction with cryopreserved allografts. During the follow-up, all patients underwent a total-body [^18^F]FDG PET/CT with subsequent radiomics analysis. In all patients, a comparative analysis between the data extracted from native aorta and cryopreserved graft for each patient was performed. Results: All patients were male with a mean age of 72.8 years (range 63–84). Mean duration of follow-up was 51.3 months (range 3–120). During the follow-up, 2 patients (16.7%) needed a redo allograft-related surgical intervention. Overall, the rate of allograft dilatation was 33.3%. No patient had a redo infection during the follow-up. Radiomics analysis showed a different signature of implanted allograft and native aorta. Comparative analysis between the native aortas and cryopreserved allografts (dilated or not) showed several statistical differences for many texture features. Conclusions: The higher metabolic activity of allografts could indicate a state of immune-mediated degeneration. This theory should be proven with prospective, multicentric studies with larger sample sizes.

## 1. Introduction

Infections of native aortas or aortic grafts are rare but potentially catastrophic life-threatening events [1,2,3]. The prevalence of mycotic aortic aneurysms ranges from 0.7% to 2.6% in the overall population with aortic dilatative disease [4]. The incidence of aortic graft infection is 0.2–6% in patients undergoing open aortic surgery [5]; on the other hand, when a primary endovascular repair was performed, the rate of endograft infection is 0.6% [6].

Management of aortic graft infection is still debated. A conservative treatment with selected antibiotic therapy is possible in selected cases [3]. However, a complete eradication of the infectious disease is possible only with the graft removal.

Surgical treatment usually consists in graft removal and flow restoring to the lower limbs [7]. Over the years, different techniques of in situ and extra-anatomical reconstructions have been proposed [8,9,10].

Early and long-term results of allograft replacement are at least similar to those of other methods to manage infrarenal aortic graft infections [10]. A recent metanalysis [11] demonstrated that cryopreserved allografts reduce the previously reported complications of the standard allografts (rupture and dilatation).

Regarding the diagnosis of aortic graft infection, the accuracy of positron emission tomography/computed tomography with [^18^F]-fludeoxyglucose (FDG-PET/CT) in a population with suspected graft infection has been previously demonstrated [12]. Radiomics analysis have been previously used in other fields of diagnostics [13,14]. It consists of a noninvasive method using machine learning to support personalized medicine. In PET imaging, very promising results concerning the ability of handcrafted features to predict the biological characteristics of lesions and to assess patient prognosis or response to treatment have been reported in the literature. PET imaging offers the promise of direct imaging of biological processes and functions.

However, no study exists in the literature about the analysis of allografts used for in situ reconstruction with [^18^F]FDG PET/CT and radiomics analysis.

The aim of this study was to evaluate the effectiveness of [^18^F]FDG PET/CT and radiomics analysis in detecting differences between the native aorta and the abdominal aortic allograft after the total eradication of infection in patients undergoing infected graft removal and in situ reconstruction with cryopreserved allografts.

## 2. Materials and Methods

Data of vascular reconstructions with allografts performed at our department were retrospectively extracted from the hospital medical records.

All patients undergoing aortic in situ reconstruction and still alive were enrolled in the present study after the total eradication of the infection; they were recalled during the first three months of 2020 before the spread of the COVID-19 pandemic.

For the purpose of this study, all patients underwent a standardized follow-up protocol including: blood exams, high-definition CT-scan of chest and abdomen, and total-body [^18^F]FDG PET/CT. A subsequent radiomics analysis was then performed.

Ethics Committee approval was claimed and obtained.

### 2.1. Cryopreserved Allograft

All aortic grafts used in the present series were cryopreserved in nitrogen liquid vapors (−140 °C) and immersed in a solution containing 80% L-Glutamine RPMI, 10% human albumin at 20%, and 10% dimethylsulfoxide. Thawing procedure was performed immediately before the surgical operation. The cryopreserved allograft was managed by suturing all minor collateral branches including intercostal, lumbar, and sacral arteries (Polypropylene 5/0) [15]. Major branches (renal or visceral arteries) were closed by a purse-string suture if necessary.

The length of the cryopreserved allograft was then calculated directly during the surgical operation. In addition, grafts were placed only after an aggressive debridement of the surgical field. When possible, an omental flap was used to cover the graft.

### 2.2. Imaging: [^18^F]FDG PET/CT and Radiomics Analysis

The administered activity of [^18^F]FDG (Gluscan, Advanced Accelerator Application, Saint-Genis-Pouilly, France) was of about 3.7 MBq/kg body weight. Patients were fasted for at least 6 h [16]. Blood glucose level was checked before the examination and a less strict criterion was accepted as recommended for infection and inflammation [17]. PET and CT images were acquired at 60–75 min after the radiopharmaceutical injection, using a PET/CT scanner (Discovery 710; General Electric Healthcare, Waukesha, WI, USA). The acquisition was performed according to the procedural recommendations of cardiac PET/CT imaging in inflammatory, infective, infiltrative, and innervation (4Is)-related cardiovascular diseases of the European Association of Nuclear Medicine (EANM) [18]. CT data were used both for the low-noise attenuation correction of PET emission data and for fusion with attenuation-corrected PET images. PET data were reconstructed iteratively by using ordered-subset expectation maximization software. Images were displayed and analyzed in axial, coronal, and sagittal planes.

Radiomics features [19] were extracted using the LIFEx software [20] after semi-automatic segmentation of the vascular segments of interest using PET VCAR software (GE Healthcare, Waukesha, WI, USA) on a General Electric workstation. A volume of interest (VOI) was drawn for each vascular segment of interest and visualized on CT images to check the anatomical correspondence.

The steps of radiomics analysis were image segmentation, image processing, feature extraction, and final feature selection/dimension reduction.

### 2.3. Statistical Analysis

All data related to the allograft procedure were retrospectively collected in a dedicated database. This included demographics, preoperative risk factors, intraoperative features, and follow-up data.

The analysis of the cohort study was retrospectively performed.

Regarding radiomics, a feature reduction was performed to avoid redundancy. Then, distribution analysis was performed. A comparative analysis between the features from native aorta and cryopreserved graft for each patient was then performed by means of the Gehan–Breslow–Wilcoxon test. Dunn’s test was used, when necessary, in case of multiple comparisons.

Continuous data were expressed as the mean ± range or median values when necessary. Categoric data were expressed as percentages.

Statistical significance was defined at the *p* < 0.05 level.

Statistical analysis was performed using SPSS software (version 24.0 for Apple; IBM Corporation, Armonk, NY, USA).

## 3. Results

Between January 2008 and December 2018, 56 vascular reconstructions with allografts were performed at our department.

This study included 12 patients undergoing abdominal aortic in situ reconstruction and still alive at the moment of the study after the total eradication of the infection. Only cryopreserved allografts have been enrolled. Figure 1 shows the clinical data.

All patients were male with a mean age of 72.8 years (range 63–84). Table 1 shows the demographic data.

The index procedures were performed for high suspicion of mycotic aortic aneurysm in 6 cases (50%), surgical aortic graft infection in 3 cases (25%), and aortic endograft infection in 3 cases (25%).

The surgical procedures performed were straight tube grafts in 5 cases (41.7%), aorto-bi-iliac grafts in 6 cases (50%), and an aorto-femoral graft in the remaining case (8.3%).

Mean duration of follow-up was 51.3 months (range 3–120).

During the follow-up period, 2 patients (16.7%) needed a redo allograft-related surgical intervention. In one case (60-month follow-up), a bilateral iliac anastomotic pseudoaneurysm was detected (Figure 2) and treated by endovascular means with the implantation of a standard endograft (AFX; Endologix Inc., Irvine, CA, USA).

In the second case (33-month follow-up), the patient, previously treated with an aorto-femoral graft, had an aortic anastomotic pseudoaneurysm treated in urgency with resection and tube Dacron graft.

Follow-up CT scans showed small not significative dilatative disease in another two patients with no indication to redo surgery.

Therefore, the overall rate of allograft dilatation was 33.3%.

### [^18^F]FDG PET/CT and Radiomics Analysis

The mean time between the surgical procedure with infected aortic graft removal and in situ reconstruction with allograft and [^18^F]FDG PET/CT was 28.7 months (range 15–66).

All [^18^F]FDG PET/CT were negative for infections during the follow-up period.

At radiomics analysis, native aortas and the cryopreserved allografts showed a different radiomic signature (Table 2).

At radiomics analysis, a dilatation of the cryopreserved allograft did not significantly affect the radiological features analyzed. On the other hand, several features showed statistical differences between native aortas and dilated cryopreserved allografts (Table 3).

## 4. Discussion

Surgical treatment in aortic graft infection consists of removal of infected tissue and/or graft and in situ or ex situ reconstruction. Many different prosthetic and biological materials have been used to perform in situ bypass in the aorto-iliac segment, but in our center, the first choice for this life-threatening condition is the cryopreserved allograft segment from brain-dead donors. Our choice is based on the higher resistance to reinfection of these grafts. We also use this kind of allografts in other complex vascular reconstructions, including massive oncological debulking, in order to avoid the use of prosthetic grafts in surgical sites at high risk of infection. In the present series, no case of reinfection was found after the removal of the infected graft and the use of a cryopreserved allograft for in situ reconstruction. In the literature, the outcomes seem to be different [21]. Overall, Couture et al. [22] reported acceptable results to treat aortic graft infections with cryopreserved allografts with few early graft-related fatal complications; in addition, long-term allograft-related complications seem to be quite common but are associated with low mortality and amputation rates, even if the recurrence of infection is 12%.

Another key point is the dilatation of the allografts during the follow-up. Fresh allografts have been mainly abandoned because of high dilatation rates over the time. Conversely, cryopreserved allografts taken from brain-dead donors have gained popularity, because they have better collagen preservation and mechanical stability and they do not affect the visco-elasticity of muscular arteries and the wall structure of elastic arteries. However, mechanisms of cryopreserved allograft degeneration include aortic wall injury and immune-related tissue damage during cryopreservation. Interestingly, mechanical injury of the crystallized allograft prior to thawing during intra-operative manipulations or during clamping of the thawed allograft is also implicated in allograft degeneration. In their recent review, Antonopoulos et al. reported a rate of allograft dilatation of about 5% with a rate of pseudoaneurysms at the anastomotic level of about 3% [11]. In the present series, the rate of allograft dilatation was 33.3%. This difference could be related to the fact that in most articles systematically analyzed in the review, no radiological examination has been performed in order to detect allograft dilatations.

Regarding mortality rates, in the systematic review [11], it was about 15% at 30 days and about 19% during the follow-up. In the present series, the mortality rates were not relevant due to the small number of patients analyzed.

Since cryopreserved allografts were useful to treat aortic infection, a deeper analysis should be performed in order to understand morphological and structural evolutions of these grafts after the surgical implantation.

Regarding allograft dilatations, the lesions could be macroscopically related to different features. Sometimes the dilatation could be concentric and not distinguishable from a native aortic aneurysm; it seems to be similar to a real aneurysmatic degeneration. In the present series, anastomotic pseudoaneurysms were reported; this could be due to some mechanical problems at suture levels in the same way it happens when an anastomosis between organic tissue and synthetic graft was performed. In the other two cases, the small dilatations detected at follow-up could be related to sutures of the lumbar arteries that could represent a point of minor resistance with progressive dilatations during the follow-up period. For this reason, it is mandatory for a meticulous preparation of the cryopreserved allograft immediately before surgical implantation in order to avoid long-term failures [15,23].

Regarding the structural and functional aspects of the cryopreserved allografts, radiomics analysis on vascular tissues is a new and innovative field [13,14,24]. Radiomics is the discipline that deals with the extraction and analysis of quantitative features from diagnostic images [14]. Radiomics is a complex process of extracting features from diagnostic images in combination with biomarkers, as reported in the white paper from the European Society of Radiology [19].

In particular, radiomics is a quantitative approach to medical imaging, which aims at enhancing the existing data available to clinicians by means of advanced mathematical analysis. The concept of radiomics, which has most broadly (but not exclusively) been applied in the field of oncology, is based on the assumption that biomedical images contain information of disease-specific processes that are imperceptible by the human eye and thus not accessible through traditional visual inspection of the generated images. Through mathematical extraction of the spatial distribution of signal intensities and pixel interrelationships, radiomics quantifies textural information by using analysis methods from the field of artificial intelligence.

Application of radiomics in vascular surgery is very limited. Charalambous et al. [25] recently published the performance of radiomics analysis in detecting aggressive endoleak type II after endovascular aortic repair.

The current limitations of radiomics analysis are the interpretability of the features, and the lack of comparison with well-established prognostic and predictive factors. In addition, radiomics analysis is usually influenced by patients’ variables including the geometry of the anatomical spaces analyzed, which have a great impact in terms of level of noise and presence of artifacts in a radiological image.

In the literature, there are no papers published about the application of this technique to analyze the allografts implanted. The main difference between cryopreserved allografts and native aorta is a higher level of [^18^F]FDG uptake in terms of SUVmin (*p* = 0.0134), SUVmax (*p* = 0.0228), and TLG (ml) (*p* = 0.0051); however, FDG uptake pattern and intensity was not a diagnostic for infection [17]. Such findings can be explained as a consequence of uptake from endothelial cells in cryopreserved allografts.

The other features that showed statistically significant differences between cryopreserved allografts (dilated or not) and native aorta belong to different categories, such as SHAPE, NGLDM (neighborhood grey-level different matrix), GLRLM (grey-level run-length matrix), and GLZLM (grey-level zone-length matrix). Overall, implanted cryopreserved allografts were more metabolically active and less homogenous than native aorta. Vessel re-endothelization or structural modifications inducing higher cellular turn-over and lower matrix homogeneity can be the reasons sustaining these findings, thus leading to either a constructive or destructive process.

Allograft degeneration is a well-known process [26]; the data found in the present series could define a low-level chronic inflammation of the cryopreserved allografts with progressive white cells infiltration that weakens the structural integrity of the allograft. Anyway, [^18^F]FDG uptake levels of native aorta and dilated allografts did not show any statistical differences. The presence of the same metabolic level of these two tissues is in contrast with the idea that dilated allografts have high inflammation levels leading to tissue degeneration. Based on the present data, the higher SUVmax and SUVmin levels could be considered a protective factor from the aneurysmatic degeneration. Higher metabolic levels and lower matrix homogeneity can, therefore, represent a constructive and not destructive process.

It is well-known that inside surgical and endovascular grafts, a new endothelium develops originating from native vessels. Our hypothesis is that a similar phenomenon could also happen in cryopreserved allografts. Therefore, the process could not stop at the endothelium level but cause a progressive degradation of the graft wall. Accepting this hypothesis, the allograft segments with higher [^18^F]FDG uptake are related to a higher metabolic activity, strictly necessary for the regeneration of the arterial wall by using the graft as a scaffold.

No data are currently available in literature to sustain this hypothesis. However, histopathological data by Plissonnier et al. [27] carefully described immunohistochemistry modifications that affect arterial allografts after implantation in rats. In these murine models, the endothelium of the allograft is subject to a rapid necrosis caused by macrophages; after that, the lumen of the allograft is covered by a monocellular layer originating from the vessel of the host with massive production of extracellular tissue. Further, Häyry et al. [26] demonstrated that between 6 and 12 months after being implanted, the tunica adventitia of the allograft seems to be fibrotic but not inflammatory, whilst media is formed by an acellular layer of compact extracellular matrix. Finally, tunica intima is constituted by a single layer of endothelial cells that covers a layer of collagen and smooth muscle cells, probably differentiated from the endothelium of the host.

It has been proposed that cellular components of the vessel wall are able to trigger the immunological reaction and, as a result, allografts should not be considered weakly antigenic. What is highlighted by these articles seems to support the theory of allograft substitution by tissue of the host. However, murine models have metabolic levels different from human ones with shorter follow-up periods; in addition, the allografts used were not previously cryopreserved. In fact, the cryopreservation process causes a complete loss of endothelial layer [28], but this seems to not affect the in vivo immune response; in humans, host humoral and cell-mediated responses are similar for both fresh and cryopreserved allografts. This inflammatory response is attenuated by immunosuppressive drugs, such as cyclosporine, exactly like other kinds of transplantation, but this seems to not change clinical outcomes and allograft evolution. In this way, the host reaction to allografts could not be considered a kind of rejection because it does not cause a structural degeneration of the allograft, but it could be interpreted like a natural adaptation and integration of the graft inside the human body [29].

Lastly, radiomics analysis showed a substantial concordance between data extracted from PET and CT images in order to support this hypothesis. However, a bigger cohort of patients is necessary to validate these outcomes. Finally, a prospective study could be useful to analyze the cryopreserved allografts before and after the implantation.

## 5. Conclusions

Cryopreserved allografts are safe and effective for treatment of infective abdominal aortic diseases.

Radiomics analysis performed on CT and PET images showed a different signature of implanted allografts and native aorta. The reason for these differences is still largely unknown. However, the higher metabolic activity of an allograft and its lower matrix homogeneity could indicate a state of immune-mediated degeneration, thus supporting the hypothesis that the allograft has a slow process of complete substitution with tissue of the host. This theory should be proven with prospective, multicentric studies with larger sample sizes.

## Figures and Tables

**Figure 1 diagnostics-12-02831-f001:**
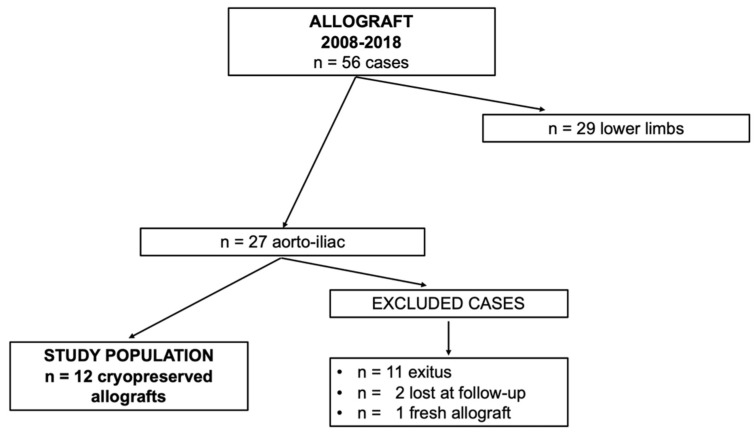
Clinical data.

**Figure 2 diagnostics-12-02831-f002:**
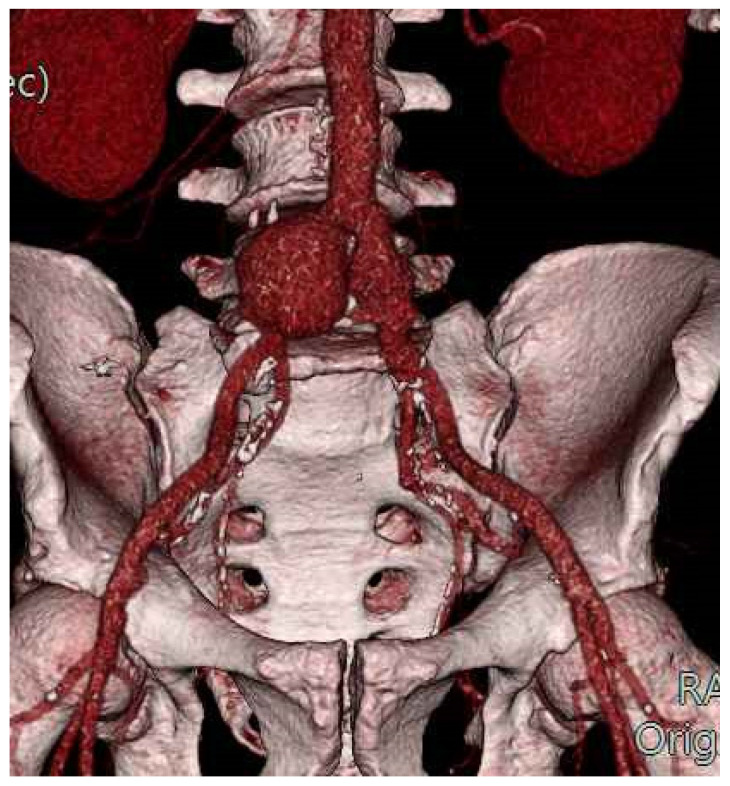
Bilateral iliac anastomotic pseudoaneurysm: volume-rendering CT-scan reconstruction.

**Table 1 diagnostics-12-02831-t001:** Demographic data.

	n = 12 Patients
Male sex	10 (83.3%)
Smoking	4 (33.3%)
Former smoking	4 (33.3%)
Hypertension	10 (83.3%)
Chronic obstructive pulmonary disease	4 (33.3%)
Diabetes mellitus	5 (41.7%)
Coronary artery disease	4 (33.3%)
Chronic renal failure *	2 (16.6%)

Continuous data are presented as the means; categorical data are given as the counts (percentage). * Glomerular filtration rate < 30 mL/min.

**Table 2 diagnostics-12-02831-t002:** Radiomics: significant differences between texture features in cryopreserved allograft vs. native aorta.

	Allograft	Native Aorta	*p*
SUVmin	0.56	0.26	0.0134
SUVmax	3.54	2.66	0.0228
TLG (mL)	51.53	34.44	0.0051
SHAPE_Volume	29.05	16.478	0.0007
SHAPE_Sphericity	0.93	0.38	0.0014
SHAPE_Compacity	2.09	1.23	0.0001
GLCM_Correlation	0.54	0.23	0.0033
GLRLM_GLNU	158.29	57.31	0.0041
GLRLM_RLNU	566.08	119.97	0.0003
NGLDM_Coarseness	0.09	<0.001	0.0041
NGLDM_Busyness	1.83	1.35	0.0192
GLZLM_GLNU	7.73	3.34	0.0008
GLZLM_ZLNU	11.69	3.56	0.0011

**Table 3 diagnostics-12-02831-t003:** Radiomics: significant differences between texture features in dilated cryopreserved allograft vs. native aorta.

	Dilated Allograft	Native Aorta	*p*
SUVmin	0.25	0.15	0.0156
SUVmax	4.18	2.59	0.0312
TLG (mL)	112.7	84.25	0.0063
SHAPE_Volume	71.27	56.48	0.0043
SHAPE_Sphericity	0.87	0.38	0.0064
SHAPE_Compacity	2.92	1.74	0.0063
GLCM_Correlation	0.62	0.23	0.0026
GLRLM_GLNU	342	259.42	0.0044
GLRLM_RLNU	1386.71	1123.42	0.0044
NGLDM_Coarseness	0.005	<0.001	0.0129
NGLDM_Busyness	3.11	2.22	0.0091
GLZLM_GLNU	14.79	10.71	0.0044
GLZLM_ZLNU	27.61	23.86	0.0452

## Data Availability

Not applicable.

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
