# Peer review of "[18F]-Fludeoxyglucose Positron Emission Tomography/Computed Tomography with Radiomics Analysis in Patients Undergoing Aortic In-Situ Reconstruction with Cryopreserved Allografts"

_diagnostics, 2022, doi:10.3390/diagnostics12112831_

Round 1

Reviewer 1 Report

Overview

It this study, authors investigated the FDG PET/CT radiomics analysis between native aorta and cryopreserved aortic graft. Although this is an interesting topic, there were critical issues in the background and methods.

Introduction

1) Background explanation and purpose are very confusing. The aim of this study was to evaluate the effectiveness of FDG PET/CT and radiomics analysis in detecting differences between the native aorta and the abdominal aortic allograft in patients undergoing infected graft removal and in-situ reconstruction with cryopreserved allograft. However, it is difficult to understand clinical value for evaluating differences in radiomics features between native aorta and aortic allograft without suspicion of infectious complication.

MATERIALS AND METHODS

2) It is necessary to describe the underlying disease and clinical situation of subjects regarding graft surgery.

3) This study was retrospective in design. It is necessary to describe clinical purpose for FDG PET/CT in these subjects.

4) It is necessary to describe the time interval between FDG PET/CT and graft surgery, which also affect the FDG uptake of graft.

5) PET and CT images were acquired consecutively 60–90 min after the radiopharmaceutical injection of using a PET/CT scanner

; The FDG uptake time is well-known to affect the PET radiomics features. Different FDG uptake time is another main limitation of this study.

6) It is necessary to describe the lesion segmentation method in detail.

7) What is the definition and extent of native aorta for radiomics analysis?

Results

8) As demographic data, it is necessary to add gender variable.

9) In the Table 2 and 3, it is necessary to show the values of each variable between cryopreserved allograft and native aorta.

Conclusion

10) It is difficult to agree with the conclusions considering the presented results and study design. Especially, post-operative inflammatory change is well-known to be the cause of FDG uptake in graft.

Author Response

Reviewer 1

Overview

It this study, authors investigated the FDG PET/CT radiomics analysis between native aorta and cryopreserved aortic graft. Although this is an interesting topic, there were critical issues in the background and methods.

Introduction

1) Background explanation and purpose are very confusing. The aim of this study was to evaluate the effectiveness of FDG PET/CT and radiomics analysis in detecting differences between the native aorta and the abdominal aortic allograft in patients undergoing infected graft removal and in-situ reconstruction with cryopreserved allograft. However, it is difficult to understand clinical value for evaluating differences in radiomics features between native aorta and aortic allograft without suspicion of infectious complication.

  1. The aim of the study is to evaluate differences in terms of radiomics analysis between the native aorta and the allograft “after the total removal of the infection”. We made a mistake into the text because the enrollment period was 2008-2018 (as reported in Figure 1… we are sorry for that). For this reason, we recalled all patients survived and free from re-infection. In fact, the purpose of the study was to analyze the differences in the native aorta and allograft recalling all patients to perform a FDG PET/CT and radiomics analysis in “a stable” clinical situation. We added some details in the Abstract and in the text.

MATERIALS AND METHODS

2) It is necessary to describe the underlying disease and clinical situation of subjects regarding graft surgery.

  1. Thanks for the comment. We already reported in the text details “The index procedures were performed for high suspicion of mycotic aortic aneurysm in 6 cases (50%), surgical aortic graft infection in 3 cases (25%), and aortic endograft in-fection in 3 cases (25%).

The surgical procedures performed were straight tube grafts in 5 cases (41.7%), aorto-bi-iliac grafts in 6 cases (50%), and aorto-femoral graft in the remaining case (8.3%).”

3) This study was retrospective in design. It is necessary to describe clinical purpose for FDG PET/CT in these subjects.

  1. Thanks for the comment. As wrote in the previous comments the purpose of FDG PET/CT is not clinical but only “scientific” in order to evaluate the differences between the native aorta and the allograft. We added a sentence in the text. We obtained the Ethics Committee approval for this study.

4) It is necessary to describe the time interval between FDG PET/CT and graft surgery, which also affect the FDG uptake of graft.

  1. Thanks for the comments. As wrote in the previous comments we recalled all patients undergoing removal of infected aortic graft and still alive at 2020. We reported in the text the mean time between the surgical procedure and the FDG PET/CT.

5) PET and CT images were acquired consecutively 60–90 min after the radiopharmaceutical injection of using a PET/CT scanner; The FDG uptake time is well-known to affect the PET radiomics features. Different FDG uptake time is another main limitation of this study.

  1. All patients were scanned using the same acquisition protocol with the scan starting 60-75 minutes after the radiopharmaceutical injection. The acquisition was performed according to the procedural recommendations of cardiac PET/CT imaging in inflammatory, infective, infiltrative, and innervation (4Is)-related cardiovascular diseases of the European Association of Nuclear Medicine (EANM) (https://doi.org/10.1007/s00259-020-05066-5) This has been added in the manuscript.

6) It is necessary to describe the lesion segmentation method in detail.

See below.

7) What is the definition and extent of native aorta for radiomics analysis?

  1. PET/CT images were visually interpreted by two experienced nuclear medicine physicians (PAE and EL) and aware of the patient’s medical history. Lesions were anatomically designed considering the CT images and FDG uptake described as absent or present. In this latter case the patter and intensity were also described. Further, a volume of interest (VOI) was draw for each vascular segment of interest on the CT images to check the anatomical correspondence.at the level of the allograft and on the upper and lower portion of the native aorta (about 5 cm length) and transposed on the PET images using the PET VCAR software (GE Healthcare, Waukesha, WI, USA) on a General Electric workstation. Then, radiomic features (n=42) were extracted from each VOI by using the LIFEx software (25) (http://www.lifexsoft.org).

Results

8) As demographic data, it is necessary to add gender variable.

  1. Thanks for the suggestion. We added the data in Table 1.

9) In the Table 2 and 3, it is necessary to show the values of each variable between cryopreserved allograft and native aorta.

  1. Thanks for the comment. We revised the tables.

Conclusion

10) It is difficult to agree with the conclusions considering the presented results and study design. Especially, post-operative inflammatory change is well-known to be the cause of FDG uptake in graft.

A. Thanks for the comments. After the revision of the article according to your suggestions we could think that the conclusions are supported but the results of the study, because we reported outcomes in patients with total eradication of the infection and with a mean distance between the procedure and the FDG PET/CT of about 30 months. For this reason, the higher metabolic activity of allograft could indicate a state of immune-mediated degeneration.

Reviewer 2 Report

I congratulate with the authors for this original study, and for their experience with cryopreserved allografts.

Following are my few observations.

Lines 28-9: you should specify your main results here in the Results section of the Abstract, not in the Conclusions section.

Lines 46-8: the possible ways of revascularizing the lower limbs are not only in-situ.

Lines: 69-70: again, these “56…” are part of the Results of your study, and should not be listed in the Materials and Methods section. In this latter, you should only say the methods of your study. That is, that you collected all the cases of vascular reconstructions with allografts performed at your Department, excluding those implanted in the lower limbs. Therefore, figure 1 belongs to the Results section.

Figure 1: these are clinical data, not demographics.

Line 191-6: please, change as follows. “In their recent review, Antonopoulos et al. reported a rate of allograft dilatation of about 5% with a rate of pseudoaneurysms at anastomotic level of about 3% [11]. In the present series the rate of allograft dilatation was 33.3%. This difference could be related to the fact that in most articles systematically analyzed in the review no radiological examination has been performed in order to detect allograft dilatations.”

Lines 199-200: “in addition, the overall population study was too much selected with strict criteria.” I don’t understand what you mean: the review from Antonopoulos is about the same cryopreserved allografts and aorto-iliac in situ reconstruction as yours. Please, explain better.

Lines 300-1: in the present form, this statement is not supported by your results.

Author Response

Reviewer 2

I congratulate with the authors for this original study, and for their experience with cryopreserved allografts.

  1. Many thanks for the comment.

Following are my few observations.

Lines 28-9: you should specify your main results here in the Results section of the Abstract, not in the Conclusions section.

  1. Thanks for the comment. We revised the text.

Lines 46-8: the possible ways of revascularizing the lower limbs are not only in-situ.

  1. Thanks for the comment. We revised the text adding the term “extra-anatomical”.

Lines: 69-70: again, these “56…” are part of the Results of your study, and should not be listed in the Materials and Methods section. In this latter, you should only say the methods of your study. That is, that you collected all the cases of vascular reconstructions with allografts performed at your Department, excluding those implanted in the lower limbs. Therefore, figure 1 belongs to the Results section.

  1. Many thanks for the comment. We moved the first part of Methods to Results. As consequence, even Figure 1 moved to Results.

Figure 1: these are clinical data, not demographics.

  1. Thanks for the comment. We changed the legend of the figure.

Line 191-6: please, change as follows. “In their recent review, Antonopoulos et al. reported a rate of allograft dilatation of about 5% with a rate of pseudoaneurysms at anastomotic level of about 3% [11]. In the present series the rate of allograft dilatation was 33.3%. This difference could be related to the fact that in most articles systematically analyzed in the review no radiological examination has been performed in order to detect allograft dilatations.”

  1. Many thanks for the comment. We changed the text as suggested.

Lines 199-200: “in addition, the overall population study was too much selected with strict criteria.” I don’t understand what you mean: the review from Antonopoulos is about the same cryopreserved allografts and aorto-iliac in situ reconstruction as yours. Please, explain better.

  1. Many thanks for the comment. We removed this sentence from the text. It is surely confounding.

Lines 300-1: in the present form, this statement is not supported by your results.

  1. We removed this sentence from the text as suggested.

Reviewer 3

Cryopreserved allograft use in infected vascular tree are well known. The follow up of patients operated on is crucial because of higher life threatening in reoccurrence of infection. The role of PET in these cases appears to be really important.  The paper seem well written and useful for publication.

A. Many thanks for the comment.

Reviewer 3 Report

Cryopreserved allograft use in infected vascular tree are well known. The follow up of patients operated on is crucial because of higher life threatening in reoccurrence of infection. The role of PET in these cases appears to be really important.  The paper seem well written and useful for publication. 
